# Mechanical Resistance of Different Implant Suprastructures: A Laboratory Study

Georgi Iliev [1], Dimitar Filtchev [1], Branka Trifković [2], Danimir Jevremović [3], Zhanina Pavlova [1], Svetoslav Slavkov [4] and Daniela Stoeva [1,*]

[1] Department of Prosthetic Dental Medicine, Faculty of Dental Medicine, Medical University, 1431 Sofia, Bulgaria; ilievdent@gmail.com (G.I.); mitko@vilem.bg (D.F.); z.pavlova@fdm.mu-sofia.bg (Z.P.)

[2] Clinic for Prosthodontics, School of Dental Medicine, University of Belgrade, 11000 Belgrade, Serbia; branka.trifkovic@stomf.bg.ac.rs

[3] School of Dentistry, University Business Academy, 26000 Pančevo, Serbia; danimir.jevremovic@sfp.rs

[4] Cranio Maxillo Facial Clinical Department, Hospital "Pirogov", 1606 Sofia, Bulgaria; slavkov1970@abv.bg

* Correspondence: stoeva.dani@gmail.com

**Abstract:** Background: Appropriate abutment selection according to the individual specificities of each patient is a leading factor in achieving high aesthetic results. Standardized titanium abutments are the most widely used due to their easy use and low cost. It is considered that customized abutments can eliminate many of the complications seen with factory abutments in prosthetic treatment. The purpose of this study is to evaluate whether customized abutments have better mechanical behavior in laboratory settings than standard ones. The null hypothesis is that customized abutments have better resistance to cyclic load and compression than factory abutments. Methods: The study model includes thirty implant suprastructure samples, fabricated digitally, divided into three groups according to the type of implant abutment and the used material: Group A (control group) comprised monolithic implant crowns made of zirconium dioxide and a titanium base; Group B (test group) comprised monolithic implant crowns made of zirconia implant crowns and a customized titanium alloy abutment; and Group C (test group) comprised monolithic implant crowns made of lithium disilicate and a customized titanium alloy abutment. The samples were subjected to dynamic load in a computer-controlled 2-axis machine that simulated masticatory movements, Chewing Simulator CS-4 (SD-Mechatronik, Westerham, Germany), for 250,000 cycles at a frequency of 2 Hz. The samples were then subjected to compressive strength testing in an Instron M 1185 universal testing machine. A metal steel disc was used as an antagonist, exerting pressure at a rate of 2 mm/min at room temperature on each sample. After conducting the laboratory tests, the samples were examined by an experienced expert under a Carl Zeiss microscope (Carl Zeiss Microscopy GmbH, Jena, Germany). Results: All samples were found to have passed the fatigue test in the masticatory simulator without any of the listed complications. The average value of the compressive strength at which the structures in each group fracture is as follows: Group A, 5669.2; Group B, 3126.5; and Group C, 1850.6. Based on the average values, it can be concluded that the combination of materials used in Group A has the greatest resistance. Conclusion: The weak link in the prosthetic complex consisting of a crown and abutment seems to be the crown. No abutment failure was found regardless of the type. However, monolithic zirconia crowns over standard titanium abutments withstand higher mechanical forces compared with zirconia and lithium disilicate crowns over customized ones. Detailed studies in clinical settings may provide more in-depth information on this issue.

**Keywords:** implants; custom abutments; mechanical resistance; fatigue loading



## 1. Introduction

The implant abutment is the connecting link between the implant platform and the crown. An implant prosthesis with a properly selected type of abutment ensures functional stability, adequate soft tissue support, and an emergence profile [1,2].

An implant prosthetic suprastructure may be a weak point due to excessive masticatory forces. Overloading can lead to abutment fracture, which prevents the implant and peri-implant complex from complications [3].

Appropriate abutment selection according to the individual specificities of each patient is a leading factor in achieving high aesthetic results [4]. Titanium abutments are defined as the "gold standard" for all areas in the mouth [5–9]. Standardized titanium abutments are the most widely used due to their easy use and low cost [10,11]. In 1996, Marchack [12] presented a clinical case report on the prosthetic restoration of an upper central incisor made with an individualized titanium abutment, in which the author emphasized the advantages of this type of restoration over standardized types. Despite efforts to unify and standardize prosthetic treatment, very often, factory solutions cannot meet the individual needs and personal requirements of each patient. In some clinical cases, a standard abutment cannot ensure the most favorable position for the future restoration and gingival architectonics [13,14]. The VAD (virtual abutment design) concept, which is a part of CAD technology, is based on the understanding that the shape of an abutment is determined by the morphology of the restoration it will support [15]. A number of authors believe that customized abutments can eliminate many of the complications seen with factory abutments in prosthetic treatment [1,12,16–18]. Fracture of implant structures is a common problem [19]. The mechanical behavior of different types of abutments in laboratory settings can provide clear guidelines for their application in clinical settings and important information regarding their advantages and disadvantages [20].

Several studies have reported that the in vitro testing of cyclic load and compressive force on an implant–prosthetic restoration complex leads to the deformation of standardized implant components and reduces their functional stability [21–23]. It is unclear whether the mechanical strengths of customized titanium alloy abutments and new CAD/CAM restorative materials have advantages compared with prosthetic restorations with standard titanium bases. The resolution of this would facilitate the choice of prosthetic components in implants and would create the conditions for more predictable and long-lasting results of prosthetics in implants. The purpose of this study is to evaluate whether customized abutments have better mechanical behavior in laboratory settings than standard ones. The null hypothesis is that customized abutments have better resistance to cyclic load and compression than factory abutments.

## 2. Materials and Methods

This study follows the experimental model presented in Figure 1.

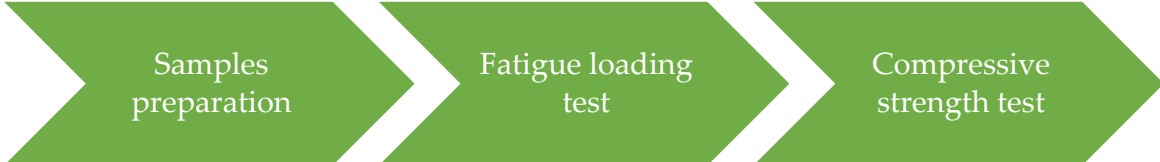

**Figure 1.** Study model.

Thirty implant suprastructure samples, divided into three groups according to the type of implant abutment and the material, were included in this study. The sample size was calculated using the software SigmaPlot 14.0 (Systat Software, Inc., San Jose, CA, USA). The sample size was calculated based on previous studies that have evaluated the mechanical performance of implant suprastructures after cyclic loading [10,24–26].

The materials used for the specimens are presented in Table 1. The work protocol for manufacturing the specimens was in line with entirely digitally based CAD/CAM technology.

**Table 1.** Materials used for the manufacture of implant abutments.

| Materials | Titanium Bases GenTek™ TiBase (Zfx, Zimmer Biomet, Warsaw, IN, USA) | Titanium Alloy Blanks GenTek™ Pre-Milled Abutment Blank (Zimmer Biomet, Warsaw, IN, USA) | Zirconium Dioxide Katana (Kuraray Noritake, Tokyo, Japan) | Lithium Disilicate IPS e.max CAD (IvoclarVivadent, Schaan, Liechtenstein) |
|---|---|---|---|---|
| Study groups | A | B, C | A, B | C |

The monolithic superstructures that were subjected to dynamic load testing were divided into 3 groups, according to the materials used, as presented in Table 2.

**Table 2.** Study groups.

| Study Group | A Control Group | | B Test Group | | C Test Group | |
|---|---|---|---|---|---|---|
| Implant suprastructure type | Monolithic implant crowns made of zirconium dioxide and titanium base | | Monolithic implant crowns made of zirconia implant crowns and customized titanium alloy abutment | | Monolithic implant crowns made of lithium disilicate and customized titanium alloy abutment | |
| Subgroup | A1—frontal | A2—distal | B1—frontal | B2—distal | C1—frontal | C2—distal |
| Quantity per subgroup | 5 | 5 | 5 | 5 | 5 | 5 |
| Quantity per group | 10 | | 10 | | 10 | |
| Total | 30 | | | | | |

Each main group was divided into two subgroups: (1) frontal (represented by the upper maxillary central incisors) and (2) distal (the first upper maxillary molars) (Figure 2).

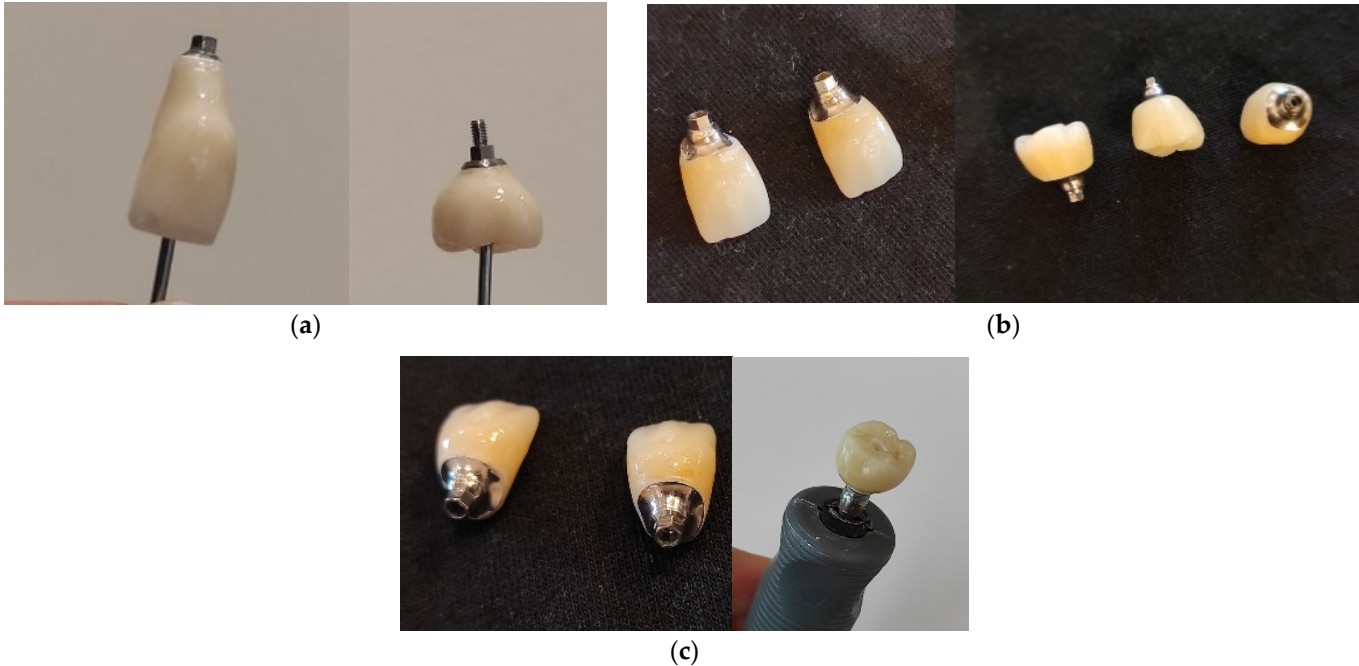

(**a**)

(**b**)

(**c**)

**Figure 2.** Study groups: (**a**) Group A; (**b**) Group B; and (**c**) Group C.

The digital protocol for making the samples had the following sequence.

A study maxillary model with installed TSV implants (Zimmer Biomet, Warsaw, IN, USA) in the area of the right central incisor and right first molar was prepared. A digital impression with an iTero Element intraoral scanner (Align Technology, San Jose, CA, USA) was taken. The file was exported into an "stl" format and thereby imported into ExoCad software (GmbH, Darmstadt, Germany) to design a monolithic zirconium crown over a GenTek titanium base (Zimmer Biomet, Warsaw, IN, USA) for Group A and a monolithic ceramic restoration over an individualized titanium superstructure for Groups B and C (Figure 3a,b). All abutments in Groups B and C were milled from customizable titanium abutments and GenTek™ Pre-milled Abutment Blanks (Zimmer Biomet, Warsaw, IN, USA) (Figure 4). The final crowns in Groups A and B were milled from Katana zirconia discs (Kuraray Noritake, Japan) in an inLab MC X5 machine (Dentsply Sirona, Charlotte, NC, USA). The crowns in Group C were made of IPS e.max CAD individual lithium disilicate blocks (IvoclarVivadent, Schaan, Liechtenstein) milled in an inLab MC XL machine (Dentsply Sirona, Charlotte, NC, USA).

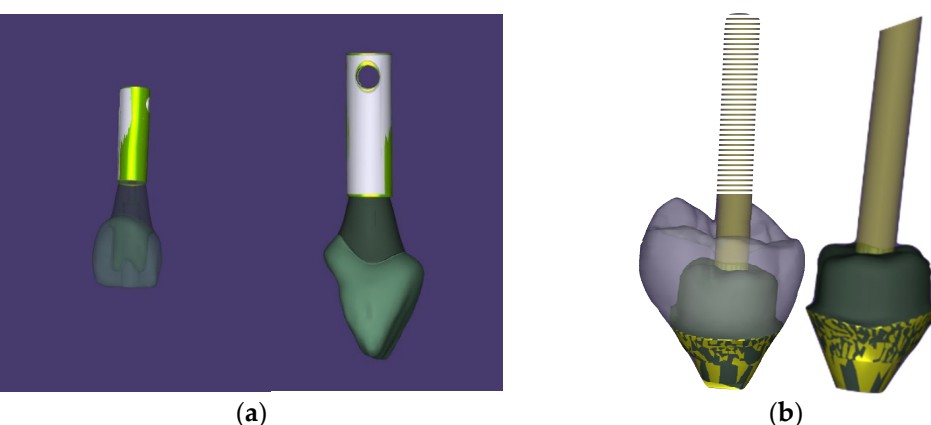

| (**a**) | (**b**) |

**Figure 3.** Digital designs of (**a**) central maxillary incisor and (**b**) upper first molar.

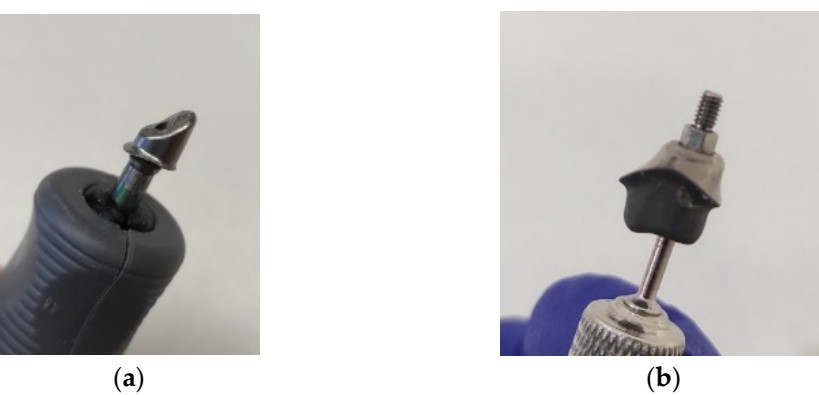

| (**a**) | (**b**) |

**Figure 4.** Milled customized titanium alloy superstructures: (**a**) central incisor and (**b**) first molar.

The sintering of the lithium disilicate monolithic crowns was performed in a Programat P300 furnace (Ivoclar Vivadent, Shaan, Liechtenstein), and that of the zirconium crowns in a Zirconmaster S furnace (VOP Ltd., Sofia, Bulgaria), followed by glazing. The cementation of the monolithic crowns in the Group A and C abutments was performed according to the manufacturer's instructions for lithium disilicate. Each sample was screw-retained over the implant and retightened after 10 min to avoid the possibility of its loosening.

The samples were stored in distilled water at 37 °C for a period of 24 h prior to testing. The implants were fixed in the holder of the fatigue testing machine immersed in polymethyl methacrylate. The samples were subjected to dynamic load in a computer-controlled 2-axis machine that simulated masticatory movements, Chewing Simulator CS-4

(SD-Mechatronik, Westerham, Germany) (Figure 5), for 250,000 cycles, representing 1 year of clinical operation of the superstructures, at a frequency of 2 Hz.

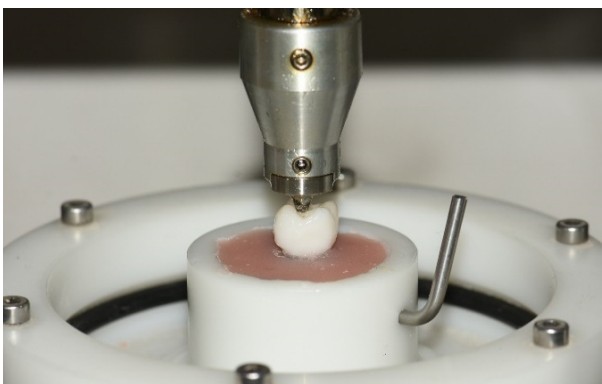

**Figure 5.** Fatigue test in a masticatory simulator CS-4 (SD-Mechatronik, Westerham, Germany).

A load of a 300 N force was applied using a standardized steel antagonist with a cone shape and 30° wall inclination, which contacted the sample at the central fissure. The vertical impact was applied according to the manufacturer's recommendations (refer to the manufacturer): upward direction, 2 mm; downward direction, 2.5 mm; upward speed, 60 mm/s; downward speed, 20 mm/s; horizontal impact, 0 mm; and speed, 20 mm/s.

The samples were then subjected to compressive strength testing in an Instron M 1185 universal testing machine. A metal steel disc was used as an antagonist, exerting pressure at a rate of 2 mm/min at room temperature on each sample. The force was applied until the sample was completely destroyed (Figure 6). The data were then tabulated using Instrument Explorer software and were subject to descriptive analysis and *t*-testing. Each sample was examined in detail under a microscope, and an examination of the type of fracture was carried out.

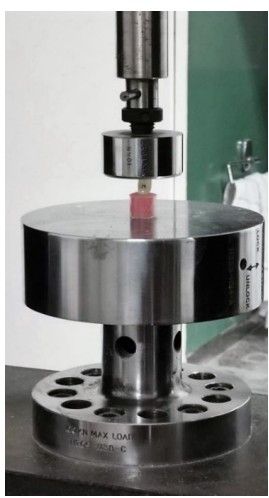
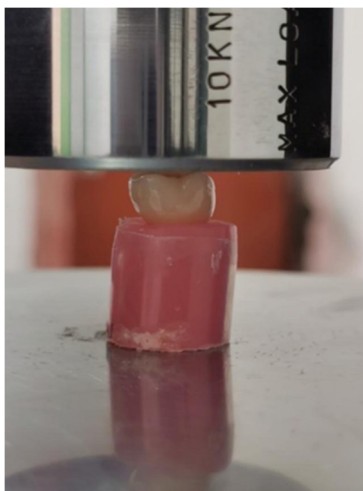

**Figure 6.** Laboratory test for compressive strength.

After conducting the laboratory tests, the samples were examined by an experienced expert under a Carl Zeiss microscope (Carl Zeiss Microscopy GmbH, Jena, Germany), and the following criteria were evaluated:

1.  Deformation of the connecting screw;
2.  Deformation of the abutment;
3.  Fracture line or crack in the monolithic crown;
4.  Visible destruction of the adhesive bond between the monolithic crown and the abutment;

5. Destruction of the monolithic crown or titanium abutment. Any defragmentation of a sample was considered destruction.

## 3. Results

A fatigue loading test measures the resistance of a material against multiple cycling loads, which cannot destroy the object separately. After conducting the laboratory fatigue strength tests, the samples were carefully examined under magnification and analyzed according to the criteria described in the task methodology. All samples were found to have passed the fatigue test in the masticatory simulator without any of the listed complications.

A compressive strength test measures the loadings that deform and destroy a material when opposite, pressing forces are applied. When the smashing force increases, the material first deforms and then fractures and is destroyed at the end.

After performing the compressive strength tests, the tested objects were subjected to repeated microscopic analysis according to the same criteria. The obtained data are summarized in Table 3.

**Table 3.** Analysis of test samples after compressive strength test.

| Tested Characteristic | Group A | Group B | Group C |
|---|---|---|---|
| Deformation of the connecting screw. | 0 | 0 | 0 |
| Deformation of the abutment. | 0 | 0 | 0 |
| Fracture line or crack in the monolithic crown. | 10 | 10 | 10 |
| Visible destruction of the adhesive bond between the monolithic crown and the abutment. | 10 | 10 | 10 |
| Destruction of the monolithic crown or titanium abutment. Any defragmentation of a sample was considered destruction. | 10 | 10 | 10 |

The microscopic analysis showed that no deformations were present in the connecting screw or the abutment after conducting both tests. In all samples, there was visible destruction of the adhesive bond and breakdown of the monolithic restoration. Two patterns of fracture behavior were demonstrated. In Group A1, the fracture was in the area of the tooth neck, horizontally and circumferentially, while in all the other subgroups, the fracture line was longitudinal and divided the permanent crown into two halves (Figure 7).

The statistical analysis compared the compressive force that caused the destruction of the suprastructures of the three study groups. Group A consisted of monolithic implant restorations made of zirconium dioxide and a titanium base. Group B consisted of monolithic zirconia implant restorations and a customized titanium alloy abutment, while Group C included monolithic lithium disilicate implant crowns and a customized titanium alloy abutment.

Each of the three groups underwent ten observations: five of the upper maxillary incisors and five of the upper maxillary first molars. Analysis of variance was performed to compare the compressive strengths of crowns from the three groups. The average value of the compressive strength at which the structures in each group fracture is as follows: Group A, 5669.2; Group B, 3126.5; and Group C, 1850.6. Based on the average values, it can be concluded that the combination of materials used in Group A has the greatest resistance. The observed differences between the 3 groups were also highly statistically significant at a standard significance level of 0.05 (F = 35.7; $p$-value = 0.000). This clearly shows the differences in the stabilities of the suprastructures in the study groups.

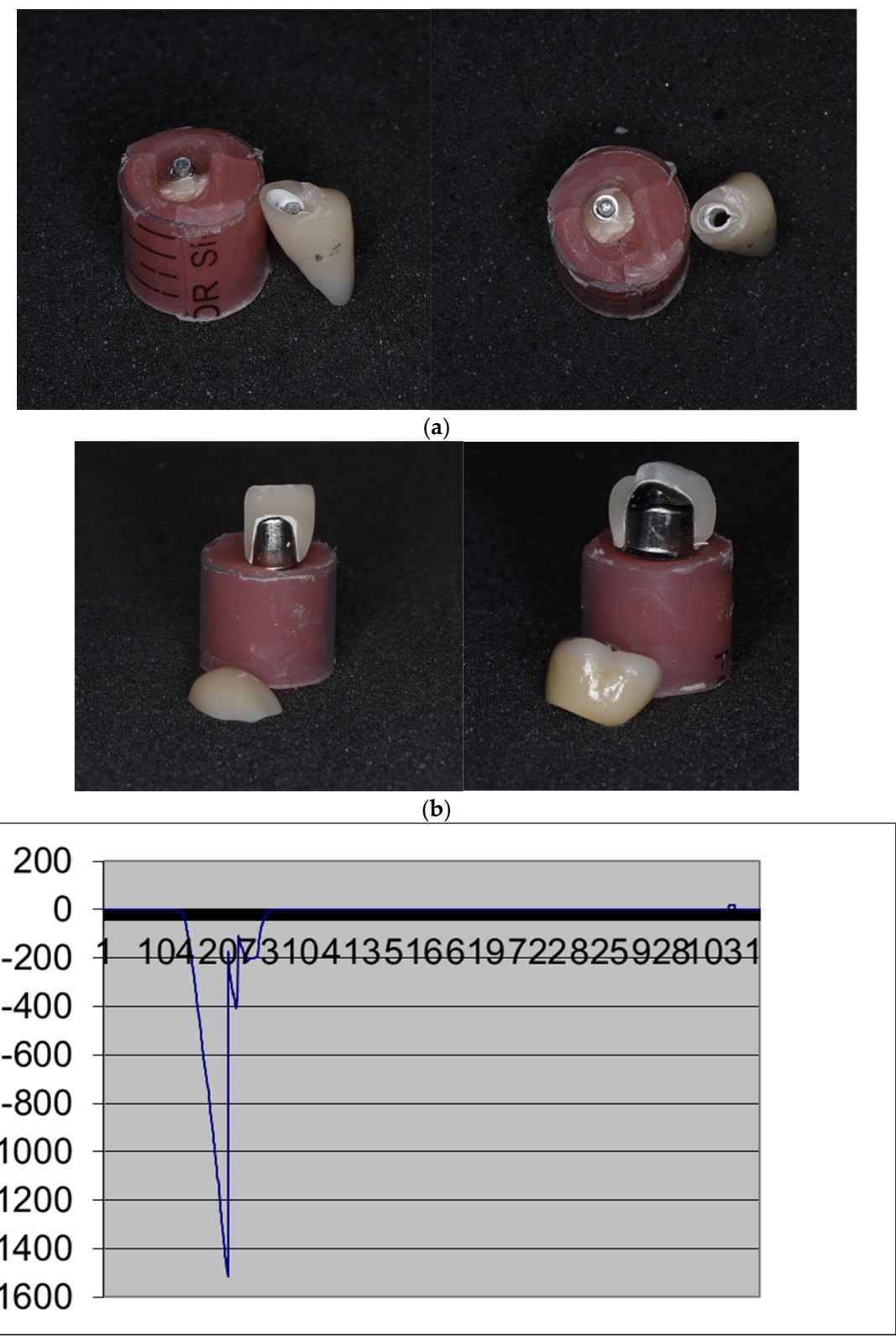

**Figure 7.** (**a**–**c**) Fracture pattern in different subgroups. (**a**) Group A1—the fracture was in the area of the tooth neck, horizontally and circumferentially. (**b**) The fracture line in all other subgroups was longitudinal and divided the permanent crown into two halves. (**c**) Diagram of fracture pattern.

As each of the three groups was composed of five incisors and molars, the statistical analysis also aimed to investigate whether there were significant differences in the resistance of the materials in each group based on the tooth type. A series of *t*-tests were carried out for each of the three groups, wherein the incisor and molar scores were directly compared. The following table shows the mean resistance values and whether there is a statistically significant difference in the resistance of the used materials depending on the tooth type. The information shows that molar-type teeth are more resistant to pressure. This resistance is statistically more significant in Groups A and B. In Group C, the difference is statistically insignificant when using a standard significance interval of 0.05 (*p*-value = 0.602) (Table 4).

**Table 4.** Comparison of the compressive strengths of implant superstructures of incisors and upper maxillary molars in each of the study groups.

|  | Group A | | Group B | | Group C | |
|---|---|---|---|---|---|---|
|  | Incisors | Molars | Incisors | Molars | Incisors | Molars |
| Mean value | 4790.2 | 6548.2 | 2217.4 | 4035.6 | 1734.4 | 1966.8 |
| *p*-value | 0.002 | | 0.009 | | 0.602 | |

The data show that the restoration resistance depends not only on the use of a specific material and abutment but also, in most cases, (except for in Group C) on whether the crown is made on an incisor or molar.

## 4. Discussion

The survival and stability of implant restorations are directly related to the biomechanical characteristics of the materials used, such as the precision between the individual components, resistance, aesthetics, etc. [27,28]. We are in agreement with Carossa et al. [3], in that implant suprastructure components prevent the implant and peri-implant tissue from overloading and severe implant complications. All samples successfully passed the fatigue test without any of the listed complications. This confirms the good mechanical behavior in laboratory settings from the data provided by Elsayed et al. [29], who obtained identical results.

After conducting compressive strength tests, it was found that there was a statistically significant difference between the test groups. The analysis of variance differentiated Group A as the one with the highest statistical difference, followed by Group B and Group C. This showed that the monolithic zirconium dioxide implant restorations on a titanium base demonstrated the highest mechanical resistance compared with the other study groups. As a probable reason, the greater thickness of the zirconia monolithic crown, which has also been stated by Denry and Kelly [30], compared with the other restoration groups, can be considered. The standardized size of the titanium base required a greater thickness of the restoration material, whereas the customized titanium abutments in the other study groups had greater volumes at the expense of the reduced thickness of the implant restorations. All samples were defragmented into two halves, with Subgroup A1's fracture being in the neck area, circumferentially and horizontally. The reason for this may be the narrower cone-shaped transmucosal part of the implant restoration in the frontal area, supported by the cylindrical titanium base. This results from the narrow and high mucogingival profile of the implants, which is determined by the specific aesthetic requirements in the visible part of the oral cavity, namely, the sufficient gingival volume, which characterizes the concept of "pink aesthetics". In the other subgroups, we observed a longitudinal fracture that separated the monolithic restoration into two parts. The high twisting forces that act in the cervical region can also be a prerequisite for fracture of the restorations in the anterior segments of the mouth. In none of the cases was the implant abutment or connecting screw deformed, which is contrary to the study data provided by Kim et al. [13] and Korsh and Walther [31]. Consequently, the applied force broke the adhesive bond between the two components of the implant suprastructure.

It is important to analyze not only the mechanical resistance of the crown—abutment complex but its components as well. Groups B and C have the same size and material of custom abutments, but the crowns are made of different materials. The received data revealed higher values in both subgroups in the tests, 2217.4 for Group B for incisors (Subgroup B1) and 4035.6 for molars (Subgroup B2). It can be concluded that the different resistance, in that case, comes from the application of different crown materials. This is why we think zirconia prosthetic constructions over individual abutments could be considered more durable in compressive strength than lithium disilicate crowns.

Likewise, the mean values in the control group, Group A, are higher in both subgroups compared with Group B. Subgroup A1 (incisors) has almost double the compressive strength resistance (4790.2) of Subgroup B1 (2217.4). The main difference between the groups comes from different abutment–titanium bases for Group A and custom abutments for Group B. In our opinion, this is the reason for the different resistance values, because both groups have the same crown material, zirconium dioxide. It should be mentioned that different abutments require different crown designs and material thicknesses, which has to be taken into consideration. However, the main focus of this study was observing some of the mechanical properties of the abutment–crown implant complexes as one, so further and more detailed experiments in this field should be conducted.

Some authors share the opinion that ceramics are hard materials and can transmit an excessively high load to a prosthetic restoration–implant complex, leading to biological or functional complications [30,31]. This calls into question the extent to which the high compressive strength of zirconia suprastructures with standard abutments is actually a positive feature.

According to other studies, implant prostheses with or without abutments have similar clinical performance after a 2-year follow-up [3]. As the authors point out, and we agree, follow-up should be after a longer period of time and different types of implant constructions should be tested for confirmation of this hypothesis.

Our results confirm this opinion regarding prosthetic implant constructions on standardized titanium bases, but according to us, the advantages of the ceramic materials that we have used over individualized superstructures are debatable. The received results confirm the null hypothesis.

The applied force on all monolithic crowns exceeded the average masticatory force in adults [32]. The results obtained have limited clinical relevance due to the number of samples and their laboratory feature, which necessitates the need for extensive in vivo studies.

It is necessary to establish whether, in the conditions of biodynamic equilibrium in the oral cavity, the studied materials would exhibit similar mechanical properties in a long-term aspect, which other authors state as well [33].

Another valuable clinical development would be the study of prosthetic restorations, a combination of standard titanium bases, and monolithic lithium disilicate crowns, which is a subject of further scientific work.

## 5. Conclusions

The development and implementation of new technologies in implantology enables the modification of implant abutments, which combined with the new methods and materials, allows for a change in the classic load protocols and the use of reliable treatment strategies. The weak link in the prosthetic complex consisting of a crown and abutment seems to be the crown. No abutment failure was found regardless of the type. However, monolithic zirconia crowns over standard titanium abutments withstand higher mechanical forces compared with zirconia and lithium disilicate crowns over customized ones. Detailed studies in clinical settings may provide more in-depth information on this issue.

**Author Contributions:** Conceptualization, D.F. and D.J.; methodology, D.S.; software, B.T.; validation, G.I., S.S. and Z.P.; formal analysis, D.F.; investigation, D.S.; resources, G.I.; data curation, D.S.; writing—original draft preparation, D.S.; writing—review and editing, G.I.; visualization, Z.P.; supervision, D.F.; project administration, D.J.; funding acquisition, S.S. All authors have read and agreed to the published version of the manuscript.

**Funding:** The materials necessary to implement this scientific research were provided under grant no. 110/24.06.2020 project.

**Data Availability Statement:** For additional or further information, please contact the corresponding author at stoeva.dani@gmail.com.

**Conflicts of Interest:** The authors declare no conflict of interest. The funders had no role in the design of this study; in the collection, analysis, or interpretation of the data; in the writing of the manuscript; or in the decision to publish the results.

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
