# Peer review of "Mechanical Resistance of Different Implant Suprastructures: A Laboratory Study"

_applsci, doi:10.3390/app13106100_

Round 1

Reviewer 1 Report

Overall its a good study. I have few minor comments

1. Please add details about whether null hypothesis was rejected or accepted 

2. Discussion is too short. Please compare the results of previously published data. Add more explanations related to the outcome of the current study. add limitations....

Author Response

Overall its a good study. I have few minor comments

1. Please add details about whether null hypothesis was rejected or accepted 

2. Discussion is too short. Please compare the results of previously published data. Add more explanations related to the outcome of the current study. add limitations....

Response to Reviewer 1:

Dear Reviewer,

Thank you very much for your kind recommendations!

1. We added details about whether the null hypothesis was accepted or rejected.
2. We made the discussion longer, by adding more details, explanations and improving the limitations information.

Kind Regards!

Reviewer 2 Report

Introduction

It is a little bit too short. Recently, some studies have hypothesized that the abutment may be a weak component of the implant-prosthodontics rehabilitation and therefore the topic of your study is very actual. Improve the introduction by adding a small paragraph on the topic to highlighted how the debate is open. For this propose discuss and cite the following recently published article 

Carossa, M.; Alovisi, M.; Crupi, A.; Ambrogio, G.; Pera, F. Full-Arch Rehabilitation Using Trans-Mucosal Tissue-Level Implants with and without Implant-Abutment Units: A Case Report. Dent. J. 202210, 116. https://doi.org/10.3390/dj10070116 

 Materials and methods

- line 84 correct ''manifacturing'' in ''manufacturing''

- Described in more details how many samples were divided between the groups and sub groups. The section started saying that 30 superstructure were selected. Then it says that they were divided between the different materials groups and then into different subgroups based on the anterior or posterior teeth. Like it is presented, it is a little bit confusing. Add after each division how many sample were considered for each group (n = X).

- For clarify the process and improve the understanding, add a flow chart of the study at the beginning of the M&M section.

- in each group there are two variables, the abutment and the crown materials. Considering that they are mixed in 3 groups, how do we know if the final results is due to the different abutments or to the different materials?

Discussion

- Discuss if the null hypothesis was accepted or rejected based on the results of the study. 

- Since this is an in vitro study, discuss a little bit how the methodology that you adopted are important to in vitro simulate the oral environment and add references supporting it. for this porpuse discuss and cite the following article doi: 10.3390/ma15041582.

Author Response

Introduction

It is a little bit too short. Recently, some studies have hypothesized that the abutment may be a weak component of the implant-prosthodontics rehabilitation and therefore the topic of your study is very actual. Improve the introduction by adding a small paragraph on the topic to highlighted how the debate is open. For this propose discuss and cite the following recently published article 

Carossa, M.; Alovisi, M.; Crupi, A.; Ambrogio, G.; Pera, F. Full-Arch Rehabilitation Using Trans-Mucosal Tissue-Level Implants with and without Implant-Abutment Units: A Case Report. Dent. J. 2022, 10, 116. https://doi.org/10.3390/dj10070116 

 Materials and methods

- line 84 correct ''manifacturing'' in ''manufacturing''

- Described in more details how many samples were divided between the groups and sub groups. The section started saying that 30 superstructure were selected. Then it says that they were divided between the different materials groups and then into different subgroups based on the anterior or posterior teeth. Like it is presented, it is a little bit confusing. Add after each division how many sample were considered for each group (n = X).

- For clarify the process and improve the understanding, add a flow chart of the study at the beginning of the M&M section.

- in each group there are two variables, the abutment and the crown materials. Considering that they are mixed in 3 groups, how do we know if the final results is due to the different abutments or to the different materials?

Discussion

- Discuss if the null hypothesis was accepted or rejected based on the results of the study. 

  • Since this is an in vitro study, discuss a little bit how the methodology that you adopted are important to in vitro simulate the oral environment and add references supporting it. for this porpuse discuss and cite the following article doi: 10.3390/ma15041582.

    Response to Reviewer 2:

    Dear Reviewer,

    Thank you very much for your constructive criticism and support!

    -We made the introduction longer and cited the articles, which you recommend.
    -We clarified the division of the groups and subgroups and added more information about that    
      in Table 2.
    -We made a flow chart in the beginning of M&M section, as you suggested, for better understanding of the study pattern.
    - We made the discussion longer, by adding more details, explanations and improving the limitations information.

    Kind Regards!

Reviewer 3 Report

Dear Authors

The presented article is a description of a typical endurance test of dental materials. The article is generally well written. However, I see one obvious flaw: the lack of a broad and clear presentation of the physical data related to the course of the destructive test of the samples: averaged data is not enough. Apart from that, I personally agree with the conclusions presented in the article, but the data should be presented more precisely.

Sincerely Yours

Reviewer

Author Response

Dear Authors

The presented article is a description of a typical endurance test of dental materials. The article is generally well written. However, I see one obvious flaw: the lack of a broad and clear presentation of the physical data related to the course of the destructive test of the samples: averaged data is not enough. Apart from that, I personally agree with the conclusions presented in the article, but the data should be presented more precisely.

Sincerely Yours

Reviewer

Response to Reviewer:

Dear Reviewer,

Thank you very much for your support and value recommendations!

We added some suplementary information for the compresive strength test for better understanding.

Kind Regards!

Round 2

Reviewer 2 Report

Dear Authors,

Thank you for addressing my points.